# Different Toxicity Profiles Predict Third Line Treatment Efficacy in Metastatic Colorectal Cancer Patients

**DOI:** 10.3390/jcm9061772

**Published:** 2020-06-07

**Authors:** Matthias Unseld, Sebastian Fischöder, Mathias Jachs, Magdalena Drimmel, Alexander Siebenhüner, Daniela Bianconi, Markus Kieler, Hannah Puhr, Christoph Minichsdorfer, Thomas Winder, Gerald W. Prager

**Affiliations:** 1Department of Medicine I, Clin. Div. of Oncology, Comprehensive Cancer Center, Medical University of Vienna, 1090 Vienna, Austria; matthis.unseld@meduniwien.ac.at (M.U.); sebastian.fischoeder@meduniwien.ac.at (S.F.); mathias.jachs@meduniwien.ac.at (M.J.); magdalena.drimmel@meduniwien.ac.at (M.D.); daniela.bianconi@meduniwien.ac.at (D.B.); markus.kieler@meduniwien.ac.at (M.K.); hannah.puhr@meduniwien.ac.at (H.P.); christoph.minichsdorfer@meduniwien.ac.at (C.M.); 2Clinic for Medical Oncology and Hematology, University Hospital Zurich and University of Zurich, 8006 Zurich, Switzerland; Alexander.Siebenhuener@usz.ch

**Keywords:** trifluridine/tipiracil, regorafenib, metastatic colorectal cancer patients, late line therapy, adverse events

## Abstract

The nucleoside trifluridine/tipiracil (TAS-102) and the multikinase inhibitor regorafenib significantly improved survival in metastatic colorectal cancer patients (mCRC). Both treatments are characterized by different treatment-related adverse events but detailed analyses of predictive side effects are rare. In this retrospective, observational, real-life study, clinical data on mCRC patients treated with trifluridine/tipiracil or regorafenib at the Medical University of Vienna, Austria and the University Hospital Zurich, Switzerland were collected. The correlation between adverse events and response or survival rates were calculated performing Fisher’s exact test and log-rank test, respectively. Common adverse events of any grade included fatigue (52%), nausea/vertigo (34%), anemia (26%), and leukopenia (22%) in trifluridine/tipiracil patients and fatigue (42%), hand-foot-skin syndrome (36%) and hoarseness (34%) in patients upon regorafenib treatment. In trifluridine/tipiracil patients the prevalence of leukopenia (*p* = 0.044) and weight loss (*p* = 0.044) was prognostic, whereas leukopenia (*p* = 0.044) and neutropenia (*p* = 0.043) predicted PFS. The disease control rate was not significantly affected. In regorafenib-treated patients, the prevalence of nausea (*p* = 0.001) was prognostic, while oral mucositis predicted PFS (*p* = 0.032) as well as the DCR (*p* = 0.039). In conclusion, we underline the efficacy of trifluridine/tipiracil and regorafenib in the real-life setting. We describe predictive adverse events like neutropenia/leukopenia, which might be used as surrogate marker in anticancer therapy beyond second line treatment.

## 1. Introduction

Colorectal cancer (CRC) is a challenging global health problem. An estimated 1.1 million new cases of CRC were diagnosed worldwide in 2018, making it the third most common type of malignancy [1]. The 5-year survival rate of CRC patients highly depends on tumor stage at the time of diagnosis, as it shows a decrease from 90% in non-metastatic patients to merely 13% in patients with metastatic colorectal cancer (mCRC) [2,3].

In recent years, new treatment options for patients with metastatic CRC have prolonged overall survival (OS) in these patients to approximately 32 months [4,5,6]. The current standard of care therapy for mCRC comprises conventional chemotherapeutical compounds as well as targeted therapy with antibodies against anti-epidermal growth factor receptor (EGFR) and vascular endothelium growth factor (VEGF) [7,8]. The implementation of these agents in standardized first-and second line treatment regimens resulted in a significant improvement in prognosis and outcome in mCRC patients. Recent advances in treatment could further improve the prognosis of mCRC patients’ options after the failure of second line treatment: 

Based on the results of the CORRECT trial, the multikinase inhibitor regorafenib has been approved for the third line treatment in patients with refractory mCRC [9]. In this particular study, regorafenib, compared to best-supportive care alone, led to a significant increase in median OS (6.4 months vs. 5.0 months, hazard ratio [HR] 0.77, *p* = 0.005) and a significant prolongation of median progression-free survival (PFS) (1.9 vs. 1.7 months, HR 0.49, *p* < 0.001). These findings were corroborated by the positive results demonstrated in the CONCUR study in an Asian population [10]. Moreover, recent studies have investigated the effects of regorafenib in other indications, leading to approval of regorafenib for the treatment of GIST and hepatocellular carcinoma [11,12,13,14].

Trifluridine/tipiracil is a recently developed oral nucleoside compound [15,16]. The efficacy of trifluridine/tipiracil in mCRC patients was investigated in the international RECOURSE trial, a phase III study comparing trifluridine/tipiracil against placebo in refractory mCRC patients [16]. The study showed positive results: patients treated with trifluridine/tipiracil showed a significant improvement in median OS (7.1 vs. 5.3 months, HR 0.58, *p* < 0.001) and median PFS (2.0 vs. 1.7 months, HR 0.48, *p* < 0.001) compared to placebo.

Considering these positive phase III trials, regorafenib and trifluridine/tipiracil offer new options in the third-line treatment of refractory mCRC. However, the most effective and safest treatment sequence in this setting remains unclear. Each agent presented a distinct toxicity profile in clinical studies. Previous studies have shown that the most common grade 3 side effects under regorafenib therapy include fatigue, hand-foot skin reactions, rash and elevation of liver enzymes [17,18]. 

Studies investigating the toxicity of trifluridine/tipiracil have found that hematological side effects of grade 3 or higher are common in trifluridine/tipiracil patients, followed by less common grade 3 side effects such as nausea and loss of appetite [17,19]. 

In summary, both drugs have shown similar effects on OS and PFS in mCRC patients, while their toxicological profile is highly different. A clinical head-to-head trial comparing regorafenib and trifluridine/tipiracil in mCRC patients is not available and analyses of the two compounds’ efficacy and side effects are scarce [20]. Treatment adherence and improved quality of life with reduced side effects was already described when regorafenib was gradually escalated in cycle 1-starting with 80 mg/day-compared to the standard dose of 160 mg/day (ReDOS) [21]. Additionally, flexible dosing showed numeric improvement on several parameters that increased tolerance, such as fatigue, hypertension, or hand-foot syndrome as shown in the REARRANGE trial [22]. Therefore, this retrospective real-life observational study aimed to investigate the efficacy and side-effects of treatment with regorafenib or trifluridine/tipiracil in mCRC patients. Moreover, we tried to elucidate the question of whether any of the reported side effects bear predictive quality for survival or disease control.

## 2. Methods

### 2.1. Study Design

The retrospective, observational, real-life study was approved by the institutional ethics committee of the Medical University of Vienna and Zurich (EC Nr.: 2189/2017) and carried out in accordance with the requirements of the International Conference on Harmonization E6 for Good Clinical Practice as laid down in the Helsinki Declaration.

### 2.2. Patients

The patients were selected from respective institutional registries, either at the Medical University of Vienna, Austria or the University Hospital Zürich, Switzerland from January 2013 to December 2017. All patients fulfilled following criteria: histologically proven adenocarcinoma of the colon or rectum with metastasis (stage IV) and an Eastern Cooperative Oncology Group (ECOG) performance status ranging from 0 to 3; pretreatment with fluoropyramidines, oxaliplatin, irinotecan, bevacizumab and, in case of RAS w.t., cetuximab or panitumumab was required, according to the label. Patients treated with trifluridine/tipiracil or regorafenib were included for statistical analysis if at least one follow up scan was performed. A total of 143 patients informed consented to one of the two treatment options, whereby 31 patients did not start treatment for different reasons (alternative treatments, deterioration of performance status, lost in follow up, disease-related events). From 112 patients who started the respective treatment, 85 patients had at least one follow-up CT scan and were considered for this retrospective analysis. Scans were performed according to respective institutional recommendations. Patients’ characteristics are displayed in Table 1. Therapy was provided upon informed consent. Further therapy algorithms, prior treatment regimens, resection and radiation, as well as tumor characteristics, were registered for patient characterization.

### 2.3. Treatment Plan and Toxicity Assessment

Trifluridine/tipiracil or regorafenib were prescribed for patients with mCRC as salvage therapy. The treatment was discontinued if the disease progressed, severe adverse events occurred or at the patient’s request. Median duration for treatment with regorafenib was 3.4 cycles, while median duration of trifluridine/tipiracil therapy was 3.2 cycles. The occurrence of adverse events was surveyed during the first cycle of application. Adverse events were described referring to CTCAE. Due to the complex analysis, the different degrees of adverse events were not considered in the evaluation.

### 2.4. Statistical Considerations

The distribution of categorical variables (e.g., localization or adverse event “yes” or “no”) was described by counts and percentages. In order to evaluate the association between different adverse events during the first cycle and the effectiveness of treatment, we performed univariate and multivariate analysis. The Fisher exact test was applied for tumor response and the log-rank test for PFS and OS. Hazard ratios or odds ratios to quantify the potential impact of an adverse event were calculated using Cox proportional-hazards regression models. Additionally, a multivariate analysis considering predefined patient background factors (age, sex, RAS mutation and localization) was performed. *p*-value < 0.05 was considered statistically significant and all conducted tests were two-sided. All calculations were carried out with IBM SPSS Statistics (version 24).

## 3. Results

### 3.1. Patient Characteristics

In this multicenter analysis, 85 patients were identified as suitable candidates between January 2013 and December 2017, as shown in Table 1. All patients had colorectal cancer with liver-, lung-, lymph node-, bone- or other metastases and have been previously treated with current standard first- and second-line therapies.

A total of 35 patients from our database were treated with regorafenib as last line therapy, while 16 patients received only trifluridine/tipiracil. Additionally, a total of 32 patients received regorafenib before trifluridine/tipiracil, whereas only two patients were treated with trifluridine/tipiracil before the initiation of regorafenib therapy.

The regorafenib and trifluridine/tipiracil cohorts comprised 69 and 50 patients, respectively. The regorafenib cohort included 49 (71%) men and 20 (29%) women, as compared to 34 (68%) men and 16 (32%) women in the trifluridine/tipiracil cohort. The median age at initial diagnosis was 60 (33–81) years in the regorafenib group and 59 (33–80) years in the trifluridine/tipiracil group. The majority of tumors, 78% of each treatment cohort, were located on the left side of the colon, including the rectum. Ras mutation status was evenly distributed between both treatment groups, whilst most of the patients of both treatment arms were BRAF wildtyp (91% and 88%). The starting dose of regorafenib was 160 mg/d in 46 (67%) patients, while 23 (33%) patients started with 120 mg or less (Table 1).

### 3.2. Toxicity Correlations

Among the 69 patients treated with regorafenib, the most frequently reported adverse event was fatigue, with 29 cases (42.0%). The second most frequent side effect, with 25 cases (36.2%), was hand-foot-skin-reaction, followed by 24 patients (34.8%) who reported hoarseness. A further 21 cases (30.4%) of weight loss or anorexia, 14 cases (20.3%) each of nausea or vertigo and hypertension, 13 cases (18.8%) of diarrhea, 11 cases (15.9%) of oral-mucositis and stomatitis, 8 cases (11.6%) each of abdominal pain and loss of appetite and 6 cases (8.7%) of paresthesia and neuropathy were reported, as shown in Table 1.

In the adverse event single observation for regorafenib, a log rank test for OS showed a significant difference (*p* < 0.001) with respect to the occurrence of nausea and vertigo vs. nausea and vertigo absent, and a log rank test for PFS showed a significant difference (*p* = 0.032) with respect to the occurrence of oral mucositis and stomatitis vs. the absence of of oral mucositis and stomatitis. With regard to disease control rate (DCR), there was a significant difference in the incidence of oral mucositis and stomatitis (exact Fischer-test: *p* = 0.039).

These differences remained even after consideration of the co-variables age, sex, Ras status and localization in a multivariate model, as shown in Table 2. 

We found that the occurrence of nausea or vertigo is significantly associated with a shorter overall survival (HR = 3.621; 95% CI: 1.519–8.630; *p* = 0.004) under regorafenib therapy. The multivariate analysis also showed that the occurrence of oral mucositis and stomatitis is a statistically significant parameter for shortened PFS (HR = 3.258; CI: 1.381–7.687; *p* = 0.007) in regorafenib therapy. Oral mucositis and stomatitis appears to be a statistically significant protective parameter in regard to the DCR (OR = 0.071; CI: 0.007–0.718; *p* = 0.0025). 

In addition, our multivariate analysis demonstrated that Ras-gene wildtype conditions were significantly associated with shorter PFS in patients that experienced oral mucositis or stomatitis (HR = 2.385; CI: 1.113–5.113; *p* = 0.025), as well as nausea or vertigo (HR = 2.197; 95% CI: 1.033–4.671; *p* = 0.041).

In the trifluridine/tipiracil treatment arm, fatigue also occurred in 26 patients (52.0%) as the most common adverse event. A total of 17 patients (34.0%) suffered from nausea and vertigo, while, in 13 cases (26.0%), anemia was recorded as the third most frequent adverse event. Another 12 cases (24.0%) of diarrhea, 11 cases (22.0%) of leukopenia, 10 cases (20.0%) of neutropenia, 9 cases (18.0%) of vomiting, 8 cases (16.0%) of weight loss or anorexia and 7 cases (14.0%) of abdominal pain were reported, as shown in Table 1.

A log rank test for PFS showed significant differences with respect to the occurrence of neutropenia (*p* = 0.043) and leukopenia (*p* = 0.010) under trifluridine/tipiracil treatment, as shown in the Kaplan–Meier curves in Figure 1A,B, respectively. Regarding OS, the occurrence of leukopenia (Figure 1C), and weight loss or anorexia, showed a significant difference in the log rank test (*p* = 0.044 for each). In respect of the a priori-defined co-variables, the differences in the PFS remained. However, the difference in OS persisted only in terms of the occurrence of weight loss or anorexia, as shown in Table 3.

Our multivariate analysis showed that the occurrence of weight loss or anorexia during trifluridine/tipiracil treatment is a statistically significant parameter (*p* = 0.022) for shortened overall survival (HR = 5.595; 95% CI: 1.286–24.344). Notably, the analysis also showed that neutropenia (HR = 0.345; 95% CI: 0.134–0.893; *p* = 0.028) and leukopenia (HR = 0.194; 95% CI: 0.066–0.575; *p* = 0.003) were significantly associated with prolonged PFS under trifluridine/tipiracil treatment. Note that febrile neutropenia was rare—only one patient (*n* = 1)—and therefore not considered in further analysis. To assess which neutrophil count bears an impact on PFS in more detail, the neutrophil counts were classified into 3 groups (< = 500, 500–1000, >1000). The results regarding PFS showed a *p*-value of (log-rank test) = 0.408. Additionally, we have analysed a possible impact of neutropenia/leucopenia on PFS/OS on the subgroups “colon” and “rectum”. The results did not show any significant differences. Next to those findings, the disease control rate did not correlate with common side effects in trifluridine/tipiracil-treated patients (Figure 1). 

## 4. Discussion

In this study, we characterize the efficacy and distinctive toxicity profile of regorafenib and trifluridine/tipiracil in the late-line treatment setting of mCRC patients. 

The safety profiles of regorafenib or trifluridine/tipiracil treatment resemble the reported adverse events of the respective randomized trials [9,16]. The most frequently reported adverse event of any grade, upon regorafenib treatment, was fatigue (42%), followed by hand-foot-skin reactions (36%), hoarseness (35%), weight loss (30%), nausea or vertigo (20%) and hypertension (20%). Notably, the frequency of weight loss and nausea or vertigo in the regorafenib cohort of any grade in our study was higher than in the CORRECT trial, while the frequency of hand-foot-skin reactions, diarrhea, oral mucositis, hypertension was lower [9]. In the trifluridine/tipiracil cohort, the most frequent adverse events of any grade were fatigue (52%), nausea or vertigo (34%), anemia (26%), diarrhea (24%), leukopenia (22%), neutropenia (20%), vomiting (18%), weight loss or anorexia (16%) and abdominal pain (14%). When compared to the RECOURSE trial, the frequency of fatigue and anemia was higher in our study, whereas the frequency of nausea, neutropenia, diarrhea, vomiting and abdominal pain was lower [16]. The frequency of weight loss or anorexia was not reported in the RECOURSE trial, however, the incidence we found in this study conforms to other previously published data [17,19].

When compared to each other, both study compounds show distinguishing toxicity profiles. While trifluridine/tipiracil commonly caused side effects of a hematological nature, regorafenib caused a higher number of adverse events. This finding was corroborated by a recently published meta-analysis [20].

Finally, we analyzed the adverse events reported under therapy with regorafenib or trifluridine/tipiracil for predictive value. We found that in the regorafenib cohort, the occurrence of nausea or vertigo was predictive for shorter overall survival (HR = 3.621, 95% CI: 1.519–8.630, *p* = 0.004). Furthermore, we were able to show in our model that the presence of oral mucositis predicted a shortened PFS (HR = 3.258, 95% CI: 1.381–7.687, *p* = 0.007), although it positively influenced the DCR (HR = 0.071, CI: 0.007–0.718, *p* = 0.025). A hypothesis for the rationale behind this notable finding remains to be further explored in future trials, but it might be speculated that mucositis more often led to treatment interruptions.

In regard to trifluridine/tipiracil, we found that weight loss or anorexia predicted a shortened OS (HR = 5.595; 95% CI: 1.286–24.344; *p* = 0.022). On the other hand, longer PFS was predicted by the occurrence of leukopenia (HR = 0.194, CI: 0.066–0.575, *p* = 0.003) and neutropenia (HR = 0.345; CI: 0.134–0.893; *p* = 0.028) in our patients and those side effects seemed to have positive effects on OS, however, this trend did not reach statistical significance in our analysis. We also investigated which neutrophil count bears an impact on PFS in more detail. Furthermore, we have analysed a possible impact of neutropenia/leucopenia on PFS/OS on the subgroups “colon” and “rectum”. However, no significant impact was determined, possibly due to the small sample size of the study. 

While the predictive quality of neutropenia for PFS in trifluridine/tipiracil therapy has already been reported in a prior trial by Hamauchi et al., our study is the first to report predictive value of leukopenia for PFS [23]. The reason for this effect still has to be elucidated. One possible explanation might be the fact that those patients who did not show better survival parameters might suffer from an increased tumor burden, leading to an increased baseline neutrophil count resulting in less neutropenia during treatment. Another explanation might be that those patients experiencing neutropenia might have been treated with the optimal dosage, while the others might be under-dosed.

Our findings underline that some of the common adverse events seen in patients treated with either one of the study drugs actually have prognostic value. These insights might have impact on the clinical routine, as they allow clinicians to evaluate certain adverse events such as neutropenia or leukopenia in patients treated with trifluridine/tipiracil in a more differentiated manner. Furthermore, our characterization of the toxic effects of the investigated drugs might facilitate the process of finding the adequate treatment for each individual mCRC patient based on their medical history, tolerances and performance status.

Our study has also limitations: Baseline pathological assessment of patients in a late line clinical trial was sometimes several years earlier, thus, no MMR status of the patients was available, which might have been interesting for its impact. Additionally, the low patient number with BRAF V600E tumors (*n* = 2) did not allow further statistical analysis on its role. The analysis was of a retrospective nature and the sample size was limited, although the two cohorts were well matched and recruited in two different centers, ensuring the high intern validity and comparability of our findings. A prospective validation of our observations is recommended. 

In conclusion, we demonstrated that both regorafenib and trifluridine/tipiracil are effective therapeutic options in the late line setting of mCRC therapy and show a distinct toxicological profile. Some of the adverse events caused by the compounds have predictive value that might support the clinician in the process of developing an individualized, patient-centered therapy in order to ensure their patient’s best possible outcome. Both regorafenib and trifluridine/tipiracil have proven their effectiveness in large, placebo-controlled trials, but in order to further investigate their toxicological profile and their optimal use in the salvage therapy of mCRC, we advocate for prospective head-to-head trials in large patient cohorts. Ultimately, the results of such trials might lead to the establishment of novel clinical practice guidelines for the salvage therapy of mCRC.

## Figures and Tables

**Figure 1 jcm-09-01772-f001:**
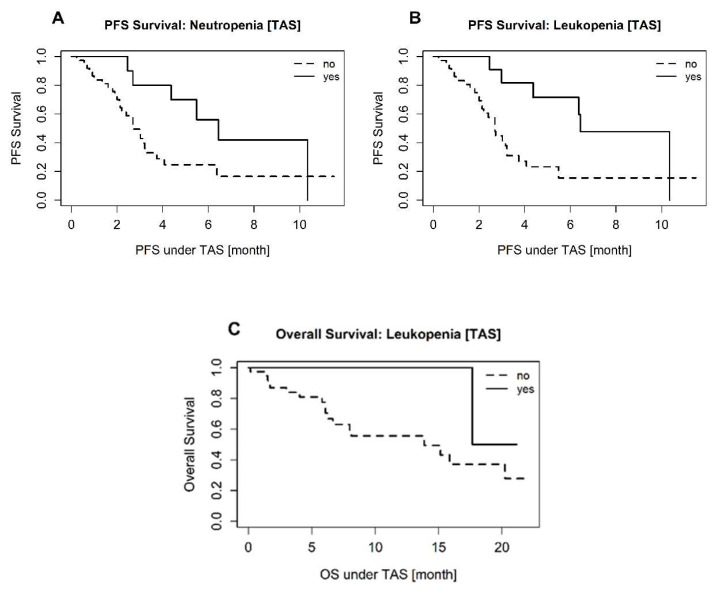
**A** + **B**: Kaplan–Meier curves showing progression-free survival under trifluridine/tipiracil. (**A**): KMC adjusted to neutropenia (yes or no) (log rank test: *p* = 0.043). (**B**): KMC adjusted to leukopenia (yes or no) (log rank test: *p* = 0.010). (**C**): Kaplan–Meier curves showing overall survival under trifluridine/tipiracil. KMC adjusted to leukopenia (yes or no) (log rank test: *p* = 0.044).

**Table 1 jcm-09-01772-t001:** Baseline patient characteristics and adverse events (CTCAE any grade).

Characteristics		Regorafenib	TAS
		*n* = 69	*n* = 50
Age at first diagnosis (years)			
Median (Range)		60 (33–81)	59 (33–80)
Sex (%)			
Male		49 (71)	34 (68)
Female		20 (29)	16 (32)
Localization (%)			
Right Colon		15 (22)	11 (22)
Left Colon + Rectum		54 (78)	39 (78)
Ras Gen Status (%)			
Wildtyp		36 (52)	25 (50)
Mutant		33 (48)	25 (50)
BRAF Status (%)			
Wildtyp		63 (91)	44 (88)
Mutant		1 (1)	1 (2)
Unknown		5 (7)	5 (10)
Starting Dose/day (%)			
80 mg	35 mg/m^2^	2 (3)	50 (100)
120 mg		21 (30)	
160 mg		46 (67)	
Therapyline (%)			
Regorafenib only		35 (51)	
Regorafenib before TAS		32 (46)	32 (64)
TAS before regorafenib		2 (3)	2 (4)
TAS only			16 (32)
Adverse events (%)			
Fatigue		29 (42.0)	26 (52.0)
Hand-Foot-Skin-Reaction		25 (36.2)	0 (0)
Hoarseness		24 (34,8)	0 (0)
Weightloss/Anorexia		21 (30.4)	8 (16)
Hypertension		14 (20.3)	0 (0)
Nausea/Vertigo		14 (20.3)	17 (34)
Diarrhea		13 (18.8)	12 (24)
Oral-Mucositis/Stomatitis		11 (15.9)	0 (0)
Abdominal Pain		8 (11.6)	7 (14)
Absence of appetite		8 (11.6)	0 (0)
Paraesthesis/Neuropathie		6 (8.7)	0 (0)
Anaemia		0 (0)	13 (26)
Neutropenia		0 (0)	10 (20)
Leukopenia		0 (0)	11 (22)
Vomiting		0 (0)	9 (18)

**Table 2 jcm-09-01772-t002:** Association between progression-free survival (PFS)/overall survival (OS)/disease control rate (DCR) and adverse events with consideration of a-priori covariates in regorafenib.

Regorafenib			
**Cox-Regression with Consideration of A-Priori Covariates**	**Nausea/Vertigo**		
	**OS**	**PFS**	**DCR**
	**HR (95% CI); *p*-Value**	**HR (95% CI); *p*-Value**	**OR (95% CI); *p*-Value**
Available Cases	*n* = 66	*n* = 53	*n* = 53
Yes vs.No	3.621 (1.519–8.630); 0.004	0.969 (0.387–2.428); 0.946	1.448 (0.361–5.809); 0.602
Age			
	0.990 (0.950–1.031); 0.626	1.017 (0.981–1.055); 0.351	0.992 (0.934–1.053); 0.786
Sex			
Female vs. Male	1.074 (0.444–2.599); 0.873	1.284 (0.531–3.109); 0.579	0.483 (0.133–1.749); 0.267
Ras-Gen-Status			
Wildtyp vs. Mutant	2.175 (0.951–4.975); 0.066	2.197 (1.033–4.671); 0.041	0.374 (0.118–1.185); 0.095
Localization			
Right vs. Left	0.845 (0.326–2.188); 0.729	1.083 (0.457–2.566); 0.857	1.043 (0.261–4.169); 0.953
**Cox-Regression with Consideration of A-priori Covariates**	**Oral-Mucositis/Stomatitis**		
	**OS**	**PFS**	**DCR**
	**HR (95% CI); *p*-Value**	**HR (95% CI); *p*-Value**	**OR (95% CI); *p*-Value**
Available Cases	*n* = 66	*n* = 53	*n* = 53
Yes vs. No	2.484 (0.924–6.677); 0.071	3.258 (1.381–7.687); 0.007	0.071 (0.007–0.718); 0.025
Age			
	0.999 (0.959–1.041); 0.956	1.026 (0.987–1.068); 0.197	0.978 (0.917–1.043); 0.493
Sex			
Female vs. Male	0.902 (0.383–2.124); 0.813	1.428 (0.591–3.450); 0.429	0.349 (0.084–1.454); 0.148
Ras-Gen-Status			
Wildtyp vs. Mutant	2.125 (0.909–4.966); 0.082	2.385 (1.113–5.113); 0.025	0.329 (0.095–1.144); 0.080
Localization			
Right vs. Left	0.835 (0.320–2.177); 0.712	1.067 (0.447–2.548); 0.884	1.029 (0.243–4.363); 0.969

OS: overall survival, PFS: progression free survival, DCR: disease control rate, HR: hazard ratio, OR: odds ratio.

**Table 3 jcm-09-01772-t003:** Association between progression-free survival (PFS)/overall survival (OS)/disease control rate (DCR) and adverse events with consideration of a-priori covariates in trifluridine/tipiracil.

TAS			
**Cox-Regression with Consideration of A-Priori Covariates**	**Weightloss/Anorexia**		
	**OS**	**PFS**	**DCR**
	**HR (95% CI); *p*-Value**	**HR (95% CI); *p*-Value**	**OR (95% CI); *p*-Value**
Available Cases	*n* = 48	*n* = 45	*n* = 45
Yes vs. No	5.595 (1.286–24.344); 0.022	0.987 (0.259–3.764); 0.985	2.982 (0.425–20.906); 0.271
Age			
	1.004 (0.962–1.049); 0.844	0.981 (0.950–1.014); 0.251	1.021 (0.957–1.090); 0.527
Sex			
Female vs. Male	2.437 (0.695–8.546); 0.164	1.319 (0.540–3.223); 0.544	0.674 (0.148–3.065); 0.610
Ras-Gen-Status			
Wildtyp vs. Mutant	1.457 (0.417–5.089); 0.555	1.192 (0.393–3.608); 0.757	1.595 (0.380–6.702); 0.524
Localization			
Right vs. Left	0.920 (0.185–4.565); 0.918	0.920 (0.185–4.565); 0.918	3.547 (0.322–39.086); 0.301
Constant			
			0.010 (-); 0.136
**Cox-Regression with Consideration of A-Priori Covariates**	**Neutropenia**		
	**OS**	**PFS**	**DCR**
	**HR (95% CI); *p*-Value**	**HR (95% CI); *p*-Value**	**OR (95% CI); *p*-Value**
Available Cases	*n* = 48	*n* = 45	*n* = 45
Yes vs. No	0.193 (0.037–1.020); 0.053	0.345 (0.134–0.893); 0.028	2.250 (0.425–11.918); 0.340
Age			
	1.018 (0.970–1.069); 0.463	0.977 (0.942–1.012); 0.189	1.007 (0.947–1.070); 0.831
Sex			
Female vs. Male	1.720 (0.579–5.107); 0.329	1.537 (0.662–3.566); 0.317	0.438 (0.108–1.784); 0.249
Ras-Gen-Status			
Wildtyp vs. Mutant	1.151 (0.340–3.890); 0.821	1.131 (0.496–2.578); 0.770	2.143 (0.467–9.827); 0.326
Localization			
Right vs. Left	1.408 (0.326–6.076); 0.646	1.212 (0.434–3.383); 0.714	4.826 (0.439–53.008); 0.198
**Cox-Regression with Consideration of A-Priori Covariates**	**Leukopenia**		
	**OS**	**PFS**	**DCR**
	**HR (95% CI); *p*-Value**	**HR (95% CI); *p*-Value**	**OR (95% CI); *p*-Value**
Available Cases	*n* = 48	*n* = 45	*n* = 45
Yes vs. No	0.138 (0.017–1.095); 0.061	0.194 (0.066–0.575); 0.003	4.634 (0.847–25.355); 0.077
Age			
	1.010 (0.966–1.056); 0.666	0.971 (0.938–1.005); 0.098	1.007 (0.944–1.073); 0.837
Sex			
Female vs. Male	1.831 (0.620–5.413); 0.274	1.663 (0.713–3.878); 0.239	0.384 (0.089–1.667); 0.201
Ras-Gen-Status			
Wildtyp vs. Mutant	1.304 (0.399–4.267); 0.661	1.401 (0.639–3.075); 0.401	2.550 (0.534–12.184); 0.241
Localization			
Right vs. Left	1.221 (0.288–5.173); 0.787	0.858 (0.320–2.296); 0.760	7.940 (0.583–108.078); 0.120

OS overall survival, PFS progression free survival, DCR disease control rate, HR hazard ratio, OR odds ratio.

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
