# Peer review of "Different Toxicity Profiles Predict Third Line Treatment Efficacy in Metastatic Colorectal Cancer Patients"

_jcm, 2020, doi:10.3390/jcm9061772_

Round 1

Reviewer 1 Report

Colorectal cancer (CRC) displays a major health burden and metastasis and disease recurrence remain challenging. In their study, the authors aimed to elucidate the toxicity profiles of the nucleoside trifluridine/tipiracil and the multikinase inhibitor regorafenib in metastatic CRC (mCRC) patients. Despite the fact that their findings on this topic will be of interest to a specialized subgroup within the field, the overall impact of this study is limited. To improve the manuscript, there are a number of minor points which should be addressed.

The authors should add a conclusion to their abstract.

In the introduction it is mentioned that “An estimated 1.4 million new cases of CRC were diagnosed worldwide in 2012”. Numbers from 2012 should be replaced with more recent data.

The labeling (A, B, C) in Figure 1 are not clearly visible and y-axes of survival curves should be labeled as “PFS Survival” or “Overall Survival”, respectively, instead of “Survival”.

Due to missing lines, the content of tables is hard to read.

Reviewer 2 Report

Unseld et al used the real world data(retrospective review) to assess clinical efficacy, side effect profiles of both regorafenib and Trifluridine-tipiracil in mCRC patients.They also assessed the prognostic values of pertinent side effect profiles; namely,  neutropenia, leukopenia, nausea, anorexia, vertigo, etc. They used only patients with colon and rectal cancers diagnosed with adenocarcinoma histology.

The work is relevant. It is the assessment of real-world experience with these 2 drugs which were used in RECOURSE trial and CORRECT trial, subsequently confirmed in other trials. Although these are not novel findings, but adds to depth of knowledge on side effects profiles.

Unseld and colleagues concluded that; nausea and vertigo decreased overall survival in regorafenib group, and oral mucositis decreased PFS in same group.

However, in Trifluridine-tipiracil group, weight loss or anorexia decreased OS, and leukopenia and Neutropenia increased PFS. Indeed, they did show that these parameters(neutropenia/leukopenia) additionally have improved OS, but not reaching statistical significance.

Major points;

-In your discussion, can you posit, as to what may be the reason why neutropenia/Leukpenia have these positive impact? e.g among those patients who did not show positive impact(PFS/OS), could there have been increased tumor burden, leading to increased baseline neutrophil count, hence less likely to have neurtopenia during treatment? or because of possible increased tumor  burden, the standard dosing might not be enough to achieve positive impact among those patients? and vice versa  for those who responded.

-for figures 1 A, 1B, as you can see, TAS-102 does not seem to have impact after 10 months of follow up on PFS. Is it that, beyond this period neutropenia/Leukopenia may not have impact? Can you demonstrate that if computationally feasible? You might use log-rank test to compare PFS between the 2 groups beyond 10 months and see if any impact.

-It will be nice to assess impact of neutropenia on PFS by using neurtophil counts  to assess at which level this positive impact is present; e.g <1000, 1500, etc. This may be clinically relevant  for readers to know at which level of neutrophils below which there is no impact, on PFS.

- Can you demonstrate what is the impact of Neutropenia/Leukopenia on colon only, and rectal only cancers, in a subgroup analysis, for OS and PFS?. 

-was there an incidence of febrile neutropenia. If present, please state the number.

Minor points;

-line 81; use "current standard therapies" instead of "state of the art".

-Table 1; frequencies for side effects should be written e.g (42.0), instead of (42,0).

-line 102; "In the adverse event single observation for regorafenib, log rank test for OS showed a significant difference (p<0.001) with respect to the occurrence of nausea and vertigo". Which is the reference here? is it " nausea and vertigo present " vs "nausea and vertigo absent" ?

-Tables 2, and 3; please check "No vs Yes". Which is baseline? I assumed "Yes Nausea/Vertigo" vs "No Nausea/Vertigo(this as baseline)".  Looking at Lines 109 onward, which you properly stated, "We found that the occurrence of nausea or vertigo is significantly associated with a shorter overall survival (HR = 3.621; 95% CI: 1.519 – 8.630; p = 0.004) under regorafenib therapy". Which seems to me Table 1 for those is;  yes Nausea/Vertigo vs no nausea/vertigo, rather than vice versa.

-indicate in tables 2 and 3, if age is continuous.

-no need to state software iteration "constant" for both tables 2 and 3

Reviewer 3 Report

The manuscript “Different toxicity profiles predict third line treatment 2 efficacy in metastatic colorectal cancer patients” submitted by Unseld and coworkers is interesting and provides accessible clinical and laboratory tools that might be helpful in monitoring patient undergoing an advance line of therapy in metastatic colorectal cancer. Even if the article is of interest, I would suggest some major revisions before acceptance. To this regard, please consider as follow.  

Major comments.

  • Introduction section – Page 2 paragraph 4. The authors discuss about toxicities of regorafenib and trifluoridin/tipiracil, which is a key topic for this manuscript. In this paragraph, I would mention studies evaluating dose escalation with regorafenib (i.e. ReDOS, REARRANGE and REGOCC studies) given its impact on adverse events (at least numerically reducing regorafenib adverse events as reported in Rearrange study).
  • Table 1 – I recommend adding to clinicopathological features the BRAF mutation assessment and the MMR status which are prognostic and predictive factors in metastatic CRC, also considering their role as per clinical guidelines (Selingmann et al. Ann Oncol, 2017; Koopman et al. Br J Cancer, 2009; Venderbosch et al. Clin Cancer Res, 2014; Innocenti et al. J Clin Oncol, 2017). Please specify accordingly in Table 1 and the manuscript as well.
  • Table 1 – Among patients receiving regorafenib, starting dose is not presented in Table 1. Did all patients start with regorafenib 160 mg or did they start from lower dosage? (i.e. 2 cp/die than escalating to 4 cp/die as per ReDOS study, or other similar studies). Please specify accordingly in Table 1 and the manuscript as well.
  • Table 1 and results – The authors describe the adverse events occurring with trifluoridine/tipiracil and regorafenib without referring to CTCAE criteria. Did they include adverse events of any grade? Please specify accordingly in the manuscript.
  • Results – I recommend, after adding BRAF and MMR status in Table 1, to consider also these two variables in the multivariate models.

Minor comments.

  • Abstract section – Page 1 line 21. I would suggest changing “using” with “performing” and to avoid the article “the” before both the types of statistical tests.
  • Abstract section – Page 1 line 30. The clinical and laboratory elements described as predictive in the manuscript acquired their role in advance line of treatment, as described by the authors. At this regard, I would add following the statement “…, which might be used as surrogate marker in anticancer therapy” also “beyond second line of treatment.”
  • Introduction section – Page 1 line 37. The authors describe colorectal cancer (CRC) epidemiology. However, the reference used by the authors dates to 2015 and more updated data on CRC stats are available. I would suggest also using the following reference: “Siegel et al. Ca Cancer J Clin, 2018”.
  • Introduction section – Page 1 line 39. The authors describe CRC survivorship. At this regard, I would also add the following reference “DeSantis et al. CA Cancer J Clin, 2014”.
  • Introduction section – Page 2 line 44. Citing the most updated CRC treatment guidelines, I would also add the following reference “Yoshino et al. Ann Oncol, 2018”.
  • Introduction section – Page 2 line 50. I would suggest removing “the application of”.
  • Introduction section – Page 2 line 62. I would suggest changing “In consideration of the respective…” with “Considering these...”.
  • Introduction section – Page 2 line 73. I would suggest adding the following reference “Abrahao et al. Clin Colorectal Can, 2018”.

Discussion – Page 8 line 187. I would suggest replacing “The findings of our study…” with “Our findings…”.

Round 2

Reviewer 1 Report

All suggestions to improve the manuscript have been addressed.

Author Response

Thank you for your statement "All suggestions to improve the manuscript have been addressed."

Reviewer 3 Report

Dear Authors, 

the authors fully addressed all issues and revised accordingly the manuscript. Indeed, the revised manuscript has been improved compared to earlier submission. However, I would recommend a further improvement before acceptance. I would recommend clarifying among the limitations, in the "Discussion" section, why BRAF assessment has not been included in the multivariate analysis (lines 234-240). 

Author Response

According to your suggestion we included following sentence in the "limitations": the  low patient number with BRAF V600E tumors (n=2) did not allow further statistical analysis on its role.